

# Analysis of influencing factors of anxiety and depression in maintenance hemodialysis patients and its correlation with BDNF, NT-3 and 5-HT levels

Xiaoyan Peng[1], Sujuan Feng[1], Poxuan Zhang[1], Shengmei Sang[2] and Yi Zhang[3]

[1] Department of Hemodialysis Center, Second Affiliated Hospital of Nantong University (First People's Hospital), Nantong, China
[2] International Guests Ward, Second Affiliated Hospital of Nantong University (First People's Hospital), Nantong, China
[3] Department of Science and Technology, Second Affiliated Hospital of Nantong University (First People's Hospital), Nantong, China

## ABSTRACT

**Objective**. The aim of this study is to examine the factors that contribute to anxiety and depression in individuals undergoing maintenance hemodialysis (MHD), as well as their association with serum levels of brain-derived neurotrophic factor (BDNF), neurotrophin-3 (NT-3), and serotonin (5-HT).

**Methods**. In May 2020 and June 2022, 120 MHD patients who received MDH treatment at our hospital were enrolled. The control group was composed of 60 healthy adults (>18) who completed the physical examination at the same time. The serum levels of BDNF, NT-3, and 5-HT in patients and clinical data of MHD patients with different degrees of anxiety and depression were compared. The Pearson correlation was used to evaluate the correlation between anxiety and depression scores and serum BDNF, NT-3,5-HT levels in patients with MHD. Multivariate analysis was employed to analyze the risk factors of anxiety and depression in MHD patients.

**Results**. The incidence of anxiety and depression in 120 MHD patients was 34.17% (41/120) and 64.17% (77/120), respectively. The levels of serum NT-3 and 5-HT in the anxiety group were higher than those in the non-anxiety and control group, and the levels of serum NT-3 in the non-anxiety group were higher than those in the control group ($P < 0.05$). The levels of serum BDNF, NT-3 and 5-HT in the depressed group were higher than those in the non-depressed group and control group, and the levels of serum NT-3 in the non-depressed group were higher than those in the control group ($P < 0.05$). SAS score was positively correlated with serum NT-3 and 5-HT levels, while the SDS score was negatively correlated with serum BDNF and positively correlated with serum NT-3 and 5-HT levels. Female, rural household registration, and restless leg syndrome were independent risk variables for anxiety in patients with MHD ($P < 0.05$). Rural household registration, economic deterioration, fatigue, insomnia, and vascular pain were independent variables of depression risk in patients with MHD.

**Conclusion**. Anxiety and depression in patients with MHD are closely related to the levels of serum BDNF, NT-3, and 5-HT. Female, rural household registration, more than eight dialysis times/month, insomnia, and restless leg syndrome are the risk factors for anxiety in patients with MHD. Rural household registration, economic deterioration, fatigue, insomnia, and vascular pain are the risk factors for depression

Corresponding author
Sujuan Feng, takeucifsj@163.com

in patients with MHD. The clinical implication of these findings suggests that these indexes may perhaps serve as biological indicators of anxiety and depression amongst patients undergoing MHD. Such investigation can hence contribute to early detection, monitoring, and potentially enable the depiction of novel therapeutic strategies for managing these adverse states.

## INTRODUCTION

Renal failure, a prevalent health condition affecting millions worldwide, refers to renal dysfunction caused by various chronic kidney diseases or heavy metals and drug exposure. It can progress to end-stage renal disease (ESRD), which affects the regular operation of various body organs and requires renal replacement therapy. Maintenance hemodialysis (MHD) is typical renal replacement therapy (*Axelsson et al., 2019*). MHD exchanges internal and external blood through a hemodialyzer to remove metabolic waste and excess water, purify the blood and maintain the acid–base balance of body fluids. The strict restriction of fluid intake during hemodialysis treatment has caused a great degree of pressure and shadow on the living habits of patients (*Hargrove et al., 2021*). Although MHD is efficient in treating ESRD patients, the treatment can largely influence the patient's quality of life, making the patients themselves worry about the disease. In addition, MHD patients need to restrict their diet and pay for financial and human resources, which increases the economic burden on their families. Psychological problems can further affect their physical health, and patients may have sleep disorders, further aggravating the psychological pressure on patients and causing anxiety and depression (*Al-Shammari et al., 2021*; *Hao et al., 2021*). Therefore, it is necessary to investigate the factors influencing anxiety and depression among maintenance hemodialysis patients.

Neurotrophic factor (BDNF) is a protein molecule that promotes neuron growth and is widely present in the cortex and hippocampus. Earlier investigations demonstrated that BDNF secretion is reduced in depressed patients (*Zhang, Yao & Hashimoto, 2016*). Neurotrophin 3 (NT-3) and BDNF belong to the same family of protein growth factors, which can maintain the survival of neurons and repair damaged neurons (*De Miranda, De Barros & Teixeira, 2020a*). 5-hydroxytryptamine (5-HT) is vital in the incidence and progression of anxiety and depression (*Yohn, Gergues & Samuels, 2017*). However, the factors influencing anxiety and depression among maintenance hemodialysis patients and their correlation with serum BDNF, NT-3, and 5-HT levels remain unknown.

Previously, BDNF, NT-3, and 5-HT have all been implicated in neuroplasticity and have been associated with neuropathological conditions such as anxiety and depression. BDNF promotes the survival of neuronal populations, NT-3 plays a key role in synaptic plasticity, and 5-HT is critically involved in mood regulation. Therefore, understanding their interplay in anxiety and depression amongst MHD patients can augment our comprehension of

the pathophysiology of these conditions, ultimately paving the way for early detection and therapeutic management. The current investigation aims to investigate the influencing factors of anxiety and depression in MHD patients and their correlation with the expression of BDNF, NT-3, and 5-HT.

## MATERIAL AND METHODS

### General information

For our study, we included MHD patients who had been on dialysis for at least three months and were aged between 18 and 75 years. We excluded patients with a history of mental disorders, severe systemic diseases, or anyone who had used neurotrophic factor-related drugs in the past three months. In addition, because certain conditions such as malignant tumors and hematological diseases could significantly influence the levels of BDNF, NT-3, and 5-HT, as well as exacerbate anxiety and depression, we excluded these patients to prevent confounding influences on our results. In addition, we set specific criteria to assess the completeness of case data. Cases were considered incomplete if records lacked crucial information such as BDNF, NT-3, 5-HT, SAS, and SDS levels. These cases were excluded to ensure the robustness and completeness of our analysis.

For our control group, to control potential selection bias, we employed stratified random sampling based on factors such as age and gender. We selected 60 healthy volunteers matching them based on age and gender with our patient group, ensuring none had a history of kidney disease or neurological disorders. Our sample size choice was informed by a power analysis, which allowed us to determine a balance between achieving statistical significance and efficient use of resources. Consequently, we selected 120 MHD patients and 60 healthy volunteers.

Anxiety and depression among MHD patients were assessed using the Zung Self-Rating Anxiety Scale and the Zung Self-Rating Depression Scale. NT-3 and 5-HT have been included as factors of interest in this study due to their recognised roles in neurotransmission and brain function (*Dunstan, Scott & Todd, 2017*), as well as their potential interplay with mood disorders like anxiety and depression. Serum levels of BDNF, NT-3, and 5-HT were measured using enzyme-linked immunosorbent assay (ELISA) kits, following the manufacturer's instructions. All samples obtained in this study were approved by the ethics committee of the Second Affiliated Hospital of Nantong University (First People's Hospital) and abided by the ethical guidelines of the Declaration of Helsinki, and ethics committee agreed to waive informed consent.

The study group consists of 120 MHD patients treated in our hospital from May 2020 to June 2022. Seventy-four males and 46 females, ranging in age from 41 to 69, with an average age of $56.41 \pm 6.92$. Our MHD patients underwent treatment thrice weekly, with each session lasting around four hours. These treatments followed exhaustive guidelines set by the National Kidney Foundation's Kidney Disease Outcomes Quality Initiative (NKF-KDOQI).

## Assessment of anxiety and depression

Self-rating Anxiety Scale (SAS) (*Dunstan & Scott, 2020*) and Self-Rating Depression Scale (SDS) (*Dunstan, Scott & Todd, 2017*) were used to evaluate depression and anxiety in both groups. The SAS and SDS have been widely used and validated for assessing anxiety and depression. The reliability, represented by Cronbach's alpha, is consistently reported to exceed 0.80 for both scales. The two scales are on a 4-point scale, including 20 items, with a total score of 80 points. SAS: no anxiety (<50 points), mild anxiety (50–60 points), moderate anxiety (61–70 points), severe anxiety (>70 points). SDS: no depression (<50 points), mild depression (50–59 points), moderate depression (60–69 points), severe depression (≥70 points).

## Collection of blood samples

Fasting venous blood samples (5 ml) were collected from patients and centrifuged at 3000 rpm for 10 min. The upper serum was collected and stored. Serum BDNF, NT-3, and 5-HT levels were determined by ELISA. Our ELISA procedures for determining BDNF, NT-3, and 5-HT levels utilized multiple controls and repeated measurements to ensure the accuracy and reliability of these tests. Furthermore, the utilized antibodies were confirmed for their specificity through pretesting.

## Observation indicators

Based on prior studies, we've identified potential risk variables including demographics (such as age and gender), disease duration, co-morbid conditions, dietary habits, and lifestyle factors. These variables are considered for their potential influence on anxiety and depression among MHD patients. Their inclusion in this study allows for a more comprehensive understanding of the complex interplay between these risk factors and the mental health problems observed in these patients.

Comparison of serum BDNF, NT-3, and 5-HT levels in patients with different anxiety levels. Comparison of BDNF, NT-3, and 5-HT levels in patients having different degrees of depression. Correlation analysis of SAS and SDS scores with serum BDNF, NT-3 and 5-HT levels in MHD patients. Correlation analysis between depression and serum BDNF, NT-3, 5-HT levels. Comparing general data between anxious and non-anxious MHD patients. Comparing general data between depressed and non-depressed MHD patients. Examination of the MHD patients' risk variables for anxiety. Examination of the MHD patients' risk variables for depression.

## Statistical methods

Before statistical analysis, we conducted thorough data preprocessing. This included data cleaning to remove incomplete cases or discrepancies. Normality checks were conducted using the Shapiro–Wilk test, and outliers identified using the z-score method were sensibly managed.

SPSS 24.0 statistical software (SPSS Inc., Chicago, IL, USA) was employed to analyze and process the data of this study. Measurement data were reported as mean ± SD ($\overline{x} \pm s$). To analyze the differences between various groups, we employed a one-way ANOVA. Given the multiple statistical tests conducted in our analysis, we adjusted the significance

**Table 1 Comparing the serum BDNF, NT-3, and 5-HT levels in patients with different degrees of anxiety ($\bar{x} \pm s$).**

| Group | BDNF (pg/ml) | NT-3 (pg/ml) | 5-HT (ng/ml) |
|---|---|---|---|
| Anxiety group ($n = 41$) | $79.02 \pm 12.82$ | $3.51 \pm 0.68^{ab}$ | $66.21 \pm 7.29^{ab}$ |
| Non-anxiety group ($n = 79$) | $79.30 \pm 10.28$ | $1.51 \pm 0.23^{a}$ | $38.90 \pm 4.25$ |
| Control group ($n = 60$) | $82.69 \pm 10.84$ | $0.24 \pm 0.08$ | $37.86 \pm 6.00$ |
| $F$ value | 1.985 | 989.211 | 382.203 |
| $P$ value | 0.140 | <0.001 | <0.001 |

Notes.
[a] represented $P < 0.05$ when compared to the control group.
[b] represented $P < 0.05$ when compared to the non-anxiety group.

threshold using the Bonferroni correction, thus appropriately managing the inflated Type I error rates. Demographic and clinical data were analyzed using Chi-square tests for categorical variables and Student's t-tests or Mann–Whitney U tests for continuous variables, according to their distribution. Count data were presented as $n$ and %, and the $c2$ test or Fisher exact test was employed to compare groups. Pearson correlation analysis was employed for analyzing the relationships between the SAS and SDS scores and serum BDNF, NT-3, and 5-HT levels. Multivariate analysis was conducted using a logistic regression model. All tests were two-sided, and a statistically significant result was $P < 0.05$. In addition to statistical significance, we calculated effect sizes (Cohen's d for group comparisons and r for correlations) for the main comparisons and correlations, to provide an understanding of the practical significance of the findings.

## RESULTS

### Anxiety and depression in MHD patients

Among the patients with MHD, the prevalence rate of anxiety was 34.17% (41/120), of which 28 cases had mild symptoms, 10 cases had moderate symptoms, and 3 cases had severe symptoms. The prevalence rate of depression was 64.17% (77/120), including 34 mild symptoms, 31 moderate symptoms, and 12 severe.

### Comparison of serum BDNF, NT-3, and 5-HT levels in patients with various degrees of anxiety

There was no significant difference in serum BDNF levels among the control, non-anxiety, and anxiety groups ($P > 0.05$). The serum levels of NT-3 and 5-HT in the anxiety group were significantly higher than in the non-anxiety and control groups. The serum NT-3 level in the non-anxiety group was better than in the control group (Table 1, Fig. 1).

### Comparison of serum BDNF, NT-3, and 5-HT levels in patients having varying degrees of depression

As shown in Table 2, the serum levels of BDNF, NT-3, and 5-HT in the depression group were significantly higher than those in the non-depression group and control group ($P < 0.05$); the serum levels of BDNF, NT-3, and 5-HT in depression group were higher than those in non-depression group and control group, and the level of serum NT-3 in the non-depression group was significantly higher than that in the control group (Fig. 2).

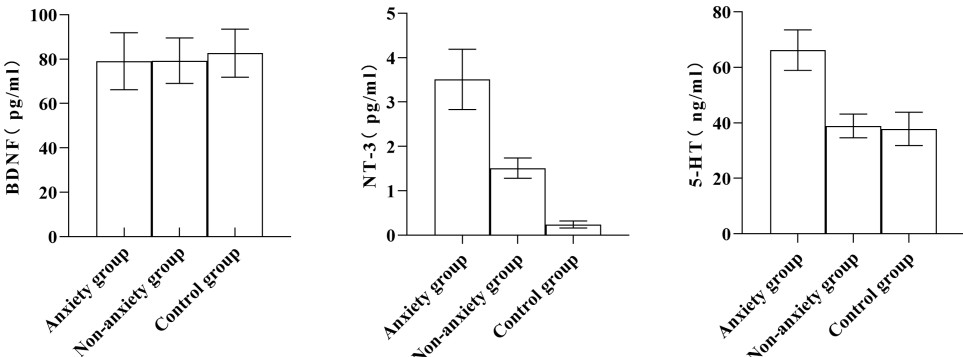

**Figure 1** Comparing the serum BDNF, NT-3, and 5-HT levels among the anxiety group, non-anxiety group, and control group.

**Table 2** Comparing the serum BDNF, NT-3, and 5-HT levels in patients having varying degrees of depression ($\bar{x} \pm s$).

| Group | BDNF (pg/ml) | NT-3 (pg/ml) | 5-HT (ng/ml) |
|---|---|---|---|
| Depression group ($n = 77$) | 76.67 ± 10.16[ac] | 2.42 ± 1.12[ac] | 51.42 ± 14.29[ac] |
| Non-depressed group ($n = 43$) | 83.76 ± 11.53 | 1.78 ± 0.76[a] | 42.53 ± 11.90 |
| Control group ($n = 60$) | 82.69 ± 10.84 | 0.24 ± 0.08 | 37.86 ± 6.00 |
| $F$ value | 8.146 | 120.852 | 24.320 |
| $P$ value | <0.001 | <0.001 | <0.001 |

Notes.
[a] represented $P < 0.05$ when compared to the control group.
[c] represented $P < 0.05$ when compared to the non-depressed group.

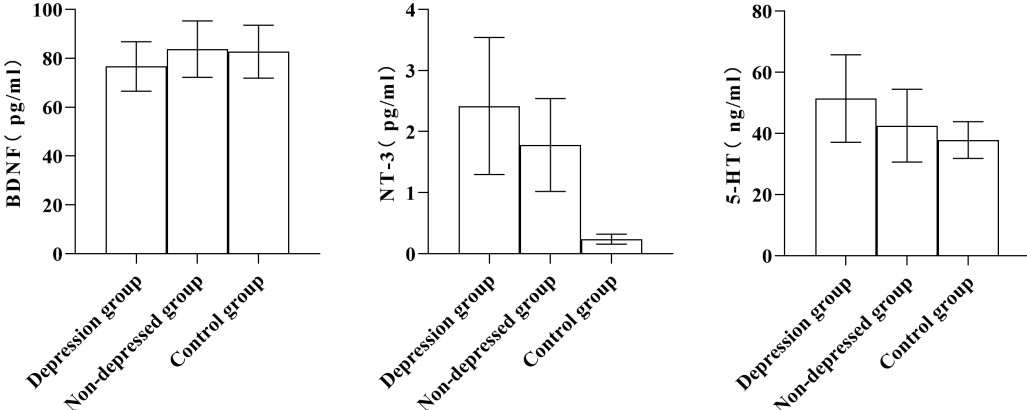

**Figure 2** Comparison of serum BDNF, NT-3, and 5-HT levels among the depression group, non-depressed group, and control group.

## Correlation analysis between anxiety and depression scores and serum BDNF, NT-3, and 5-HT levels in MHD patients

In our data, we note that serum NT-3 and 5-HT levels were found to positively correlate with the SAS score, interrelating elevated levels of these markers with increased anxiety

**Table 3** Correlation analysis between anxiety and depression scores and serum BDNF, NT-3, and 5-HT levels in MHD patients ($\bar{x} \pm s$).

| Group | BDNF | | NT-3 | | 5-HT | |
|---|---|---|---|---|---|---|
| | r value | P value | r value | P value | r value | P value |
| SAS score | −0.048 | 0.600 | 0.834 | <0.001 | 0.826 | <0.001 |
| SDS score | −0.242 | 0.008 | 0.255 | 0.005 | 0.287 | 0.002 |

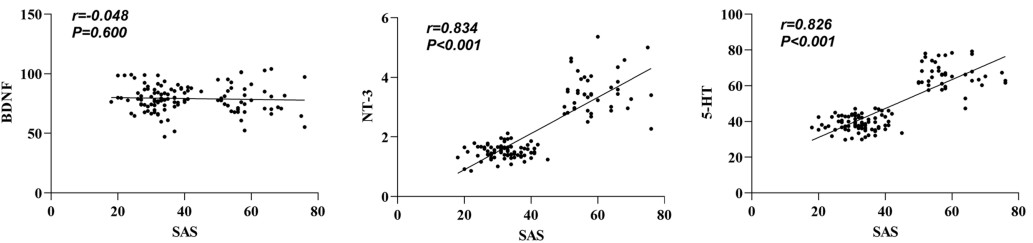

**Figure 3** Correlation between SAS score and serum BDNF, NT-3, and 5-HT levels.

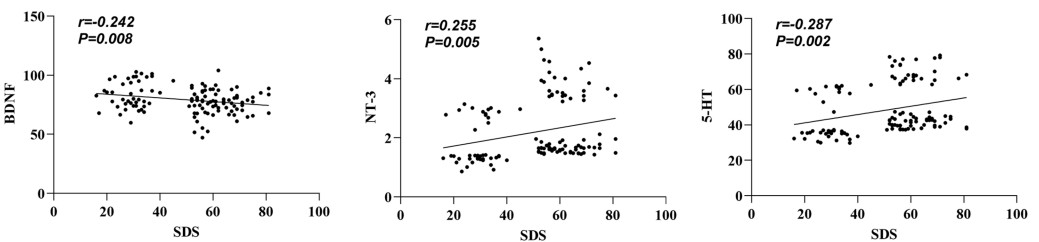

**Figure 4** Correlation between SDS score and serum BDNF, NT-3, and 5-HT levels.

symptoms. Conversely, it was observed that an increased serum BDNF level negatively correlated with the SDS score, implicating higher BDNF concentration with alleviated depressive symptoms (Table 3, Figs. 3–4). In addition, a positive correlation existed between the SDS score and NT-3 and 5-HT serum levels ($P < 0.05$).

## Univariate analysis of anxiety-related factors in MHD patients
The proportion of females, rural residents, monthly dialysis times >8 times, insomnia, and restless legs in the anxiety group was more significant than those in the non-anxiety group, and the differences were statistically significant ($P < 0.05$). No significant differences were observed in age, education level, marital status, smoking, drinking, working status, economic status, fatigue, and vascular pain between both groups ($P > 0.05$, Table 4).

## Multivariate regression analysis of anxiety-related factors in MHD patients
Anxiety in MHD patients was taken as the dependent variable (anxiety = 1, non-anxiety = 0), and factors with statistically significant differences in univariate analysis (gender, household registration, monthly dialysis times, insomnia, restless leg syndrome) were

**Table 4** Univariate analysis of anxiety-related factors in hemodialysis patients (cases (%)).

| Item | Anxiety group ($n = 41$) | Non-anxiety group ($n = 79$) | $\chi^2$ value | P value/Fisher exact probability value |
|---|---|---|---|---|
| Gender | | | | |
| Male | 18 (43.90) | 56 (70.89) | 8.314 | 0.004 |
| Female | 23 (56.10) | 23 (29.11) | | |
| Age | | | | |
| ≤60 years old | 30 (73.17) | 50 (63.29) | 1.186 | 0.276 |
| >60 years old | 11 (26.83) | 29 (36.71) | | |
| Educational level | | | | |
| Below high school | 20 (48.78) | 36 (45.57) | 0.112 | 0.738 |
| High school and above | 21 (51.22) | 43 (54.43) | | |
| Marital status | | | | |
| Married | 38 (92.68) | 65 (82.28) | 2.403 | 0.121 |
| Single (divorced or widowed) | 3 (7.32) | 14 (17.72) | | |
| Household registration | | | | |
| Urban | 23 (56.10) | 64 (81.01) | 8.404 | 0.004 |
| Rural | 18 (43.90) | 15 (18.99) | | |
| Smoking | | | | |
| Yes | 18 (43.90) | 44 (55.70) | 1.503 | 0.220 |
| No | 23 (56.10) | 35 (44.30) | | |
| Drinking | | | | |
| Yes | 1 (2.40) | 6 (7.59) | – | 0.420 |
| No | 40 (97.56) | 73 (92.41) | | |
| Working status | | | | |
| In-service | 5 (12.20) | 9 (11.39) | – | 1.000 |
| Unemployed | 36 (87.80) | 70 (88.61) | | |
| Monthly dialysis times | | | | |
| ≤8 times | 4 (9.76) | 32 (40.51) | 12.154 | <0.001 |
| >8 times | 37 (90.24) | 47 (59.49) | | |
| Economic status | | | | |
| General or good | 18 (43.90) | 40 (50.63) | 0.490 | 0.484 |
| Worsening | 23 (56.10) | 39 (49.37) | | |
| Fatigue | | | | |
| Yes | 18 (43.90) | 37 (46.84) | 0.094 | 0.760 |
| No | 23 (56.10) | 42 (53.16) | | |
| Insomnia | | | | |
| Yes | 26 (63.41) | 30 (37.97) | 7.019 | 0.008 |
| No | 15 (36.59) | 49 (62.03) | | |
| Vascular pain | | | | |
| Yes | 23 (56.10) | 46 (58.23) | 0.050 | 0.823 |
| No | 18 (43.90) | 33 (41.77) | | |
| Restless legs syndrome | | | | |

**Table 4** (*continued*)

| Item | Anxiety group (n = 41) | Non-anxiety group (n = 79) | χ² value | P value/Fisher exact probability value |
|---|---|---|---|---|
| Yes | 40 (97.56) | 25 (31.65) | 47.238 | <0.001 |
| No | 1 (2.44) | 54 (68.35) | | |

**Table 5  Multivariate logistic regression analysis of anxiety-related factors in MHD patients.**

| Factor | B value | SE | Wald value | P value | OR value | 95% CI |
|---|---|---|---|---|---|---|
| Gender | 1.648 | 0.668 | 6.090 | 0.014 | 5.198 | 1.404~19.249 |
| Household registration | 2.117 | 0.876 | 5.837 | 0.016 | 8.310 | 1.491~46.303 |
| Insomnia | 1.030 | 0.635 | 2.634 | 0.105 | 2.801 | 0.807~9.714 |
| Monthly dialysis times | 1.403 | 0.723 | 3.768 | 0.052 | 4.066 | 0.986~16.759 |
| Restless legs syndrome | 5.431 | 1.241 | 19.149 | <0.001 | 228.472 | 20.060~2602.147 |
| Constant | | | | | | |

taken as independent variables. The values were assigned, gender (male = 0, female = 1), household registration (urban = 0, rural = 1), insomnia (no = 0, yes = 1), monthly dialysis times ($\leq$8 times = 0, >8 times = 1), restless legs syndrome (no = 0, yes = 1), and they were incorporated into the Logistic regression model. The outcomes revealed that female, rural household registration, and restless legs syndrome were independent variables of risk for anxiety in MHD patients ($P < 0.05$, Table 5). The odds ratios and their 95% confidence intervals for each independent variable in the logistic regression model have now been included, to better understand the impact of these variables on the likelihood of experiencing anxiety amongst MHD patients.

## Univariate analysis of depression-related factors in MHD patients

The proportions of a single person, rural household registration, worsening economic status, fatigue, insomnia, and vascular pain in the depression group were more significant than those in the non-depressed group, with statistically significant differences ($P < 0.05$). The two groups did not significantly differ in sex, age, education, smoking, drinking, working status, monthly dialysis times, and restless leg syndrome ($P > 0.05$, Table 6).

## Multivariate analysis of depression-related factors in MHD patients

Whether MHD patients were depressed as the dependent variable (depression = 1, non-depression = 0), factors with statistically significant differences (marital status, household registration, economic status, fatigue, insomnia, vascular pain) in univariate analysis were taken as independent variables. The values were assigned, marital status (married = 0, single = 1), household registration status(urban = 0, rural = 1), economic status (general or good = 0, worsening = 1), fatigue (no = 0, Yes = 1), insomnia (no = 0, yes = 1), vascular pain (no = 0, yes = 1), and they were incorporated in the logistic regression model. The results showed that rural household registration, economic deterioration, fatigue, insomnia, and vascular pain were independent variables of depression risk in patients with MHD (Table 7). The odds ratios and their 95% confidence intervals for each

**Table 6** Univariate analysis of depression-related factors in hemodialysis patients (cases (%)).

| Item | Depression group (*n* = 77) | Non-depressed group (*n* = 43) | $\chi^2$ value | *P* value |
|---|---|---|---|---|
| Gender | | | | |
| Male | 48 (62.34) | 26 (60.47) | 0.041 | 0.840 |
| Female | 29 (37.66) | 17 (39.53) | | |
| Age | | | | |
| ≤60 years old | 49 (63.64) | 31 (72.09) | 0.888 | 0.346 |
| >60 years old | 28 (36.36) | 12 (27.91) | | |
| Educational level | | | | |
| Below high school | 35 (45.45) | 21 (48.84) | 0.127 | 0.722 |
| High school and above | 42 (54.55) | 22 (51.16) | | |
| Marital status | | | | |
| Married | 61 (79.22) | 42 (97.67) | 7.727 | 0.005 |
| Single (divorced or widowed) | 16 (20.78) | 1 (2.33) | | |
| Household registration | | | | |
| Urban | 48 (62.34) | 39 (90.70) | 11.131 | 0.001 |
| Rural | 29 (37.66) | 4 (9.30) | | |
| Smoking | | | | |
| Yes | 38 (49.35) | 24 (55.81) | 0.462 | 0.497 |
| No | 39 (50.65) | 19 (44.19) | | |
| Drinking | | | | |
| Yes | 4 (5.19) | 3 (6.98) | – | 0.700 |
| No | 73 (94.81) | 40 (93.02) | | |
| Working status | | | | |
| In-service | 8 (10.39) | 6 (13.95) | 0.340 | 0.560 |
| Unemployed | 69 (89.61) | 37 (86.05) | | |
| Monthly dialysis times | | | | |
| ≤8 times | 20 (25.97) | 16 (37.21) | 1.659 | 0.198 |
| >8 times | 57 (74.03) | 27 (62.79) | | |
| Economic status | | | | |
| General or good | 27 (35.06) | 31 (72.09) | 15.149 | <0.001 |
| Worsening | 50 (64.94) | 12 (27.91) | | |
| Fatigue | | | | |
| Yes | 42 (54.55) | 13 (30.23) | 6.570 | 0.010 |
| No | 35 (45.45) | 30 (69.77) | | |
| Insomnia | | | | |
| Yes | 47 (61.04) | 9 (20.93) | 17.834 | <0.001 |
| No | 30 (38.96) | 34 (79.07) | | |
| Vascular pain | | | | |
| Yes | 50 (64.94) | 19 (44.19) | 4.861 | 0.027 |
| No | 27 (35.06) | 24 (55.81) | | |
| Restless legs syndrome | | | | |

| Item | Depression group (*n* = 77) | Non-depressed group (*n* = 43) | $\chi^2$ value | *P* value |
|------|------|------|------|------|
| Yes | 41 (53.25) | 24 (55.81) | 0.073 | 0.787 |
| No | 36 (46.75) | 19 (44.19) | | |

**Table 7  Multivariate logistic regression analysis of depression-related factors in MHD patients.**

| Factor | $\beta$ value | SE | Wald value | *P* value | OR value | 95% CI |
|------|------|------|------|------|------|------|
| Marital status | 1.854 | 1.142 | 2.635 | 0.105 | 6.387 | 0.681~59.925 |
| Household Registration status | 2.391 | 0.752 | 10.110 | 0.001 | 10.929 | 2.503~47.731 |
| Economic status | 1.717 | 0.556 | 9.537 | 0.002 | 5.570 | 1.873~16.564 |
| Fatigue | 1.721 | 0.573 | 9.035 | 0.003 | 5.591 | 1.820~17.177 |
| Insomnia | 2.137 | 0.588 | 13.197 | <0.001 | 8.471 | 2.675~26.827 |
| Vascular pain | 1.639 | 0.572 | 8.212 | 0.004 | 5.151 | 1.679~15.805 |
| Constant | −3.325 | 0.766 | 18.836 | <0.001 | 0.036 | |

independent variable in the logistic regression model have now been included, to better understand the impact of these variables on the likelihood of experiencing anxiety amongst MHD patients.

## DISCUSSION

The prognosis of ESRD patients can be significantly improved by MHD treatment, but patients will still have some complications; long-term treatment and economic pressure may induce anxiety, depression, and other adverse emotions. According to relevant studies, adverse emotions can seriously affect the prognosis of patients, increase the risk of adverse events, and even increase the mortality of patients. Hence, analyzing the influencing factors of anxiety and depression in MHD patients is necessary (*Meng et al., 2022*; *Schouten et al., 2019*). Many studies have shown the connection between serum BDNF levels and anxiety and depression (*Rahmani, Rahmani & Rezaei, 2020*; *Notaras & Van den Buuse, 2020*). As per the relevant research, BDNF is strongly related to nerve cell growth, differentiation, and plasticity (*Cagni et al., 2017*; *Yang et al., 2020*). NT-3 is synthesized and released by astrocytes, which can maintain the normal functional state of neurons. Its distribution is wide in the central and peripheral nervous systems and participates in neurons' differentiation and damage repair. Earlier research demonstrates that its level is strongly linked to depression (*De Miranda, De Barros & Teixeira, 2020b*; *Akbaba et al., 2018*). 5-HT is widely distributed in brain nerves and the pineal gland, which belongs to indoleamine. 5-HT can participate in biological functions, including appetite, memory, emotion regulation, sleep, and stress response. 5-HT neurotransmitters in brain regions can induce depression and anxiety. In addition to examining the relationships between anxiety and depression in MHD patients and serum levels of BDNF, NT-3, and 5-HT, this research sought to determine the factors that may influence these conditions. In the current research, 120 MHD patients were chosen for the study to analyze their anxiety
and depression. The incidence of anxiety in MHD patients was 34.17% (41/120), including 28 mild cases, 10 moderate cases, and three severe cases; the incidence of depression was 64.17% (77/120), including 34 mild cases, 31 moderate cases, and 12 severe cases. The incidence of anxiety and depression is in line with the earlier reported research (*Fleishman, Dreiher & Shvartzman, 2020*; *Ma et al., 2021*). Our sample size selection, based on a power analysis, enables a statistically sound representation within our population. This increased the generalizability of our findings to the broader context of MHD patients and healthy individuals.

In the present investigation, the serum levels of NT-3 and 5-HT in the anxiety group were significantly different from those in the non-anxiety and control groups.

Serum NT-3 and 5-HT levels were more significant in the anxiety group than in the non-anxiety and control groups. Serum NT-3 levels were higher in the non-anxiety group than in the control group ($P < 0.05$). The serum levels of BDNF, NT-3, and 5-HT in the depression group were higher than those of the non-depressed group and the control group, and the serum levels of NT-3 in the non-depression group were more remarkable than that of the control group ($P < 0.05$). The serum NT-3 and 5-HT levels and the SAS score were significantly correlated, while the serum BDNF level and the SDS score had a negative correlation. In contrast, a positive correlation existed between the SDS score and NT-3 and 5-HT serum levels ($P < 0.05$). These results indicated that serum BDNF, NT-3, and 5-HT levels were closely related to MHD patients' anxiety and depression. BDNF and NT-3 are both cerebral neurotrophic factors that can regulate the central nervous system. BDNF can inhibit neuronal death and promote axonal repair and regeneration (*Yang et al., 2016*). NT-3 is widely distributed in the skin, muscle, and central nervous system and can maintain and repair synaptic structure (*Pallavi et al., 2016*). It is congruent with the findings of this study that serum 5-HT is closely associated with affective disorders in MHD patients, according to earlier research (*García-Gutiérrez et al., 2020*).

Anxiety and depression are common mental symptoms of MHD patients, and this bad psychological state can affect the immune function and nutritional status of patients to a large extent and affect the patient's prognosis. In this study, the anxiety group had a higher proportion of females, rural household registration, monthly dialysis times >8 times, insomnia, and restless legs syndrome than the non-anxiety group, with statistically significant differences ($P < 0.05$). Logistic regression analysis demonstrated that female gender, rural household registration, and restless legs syndrome were independent variables of risk for anxiety in MHD patients ($P < 0.05$). The occurrence of anxiety in female patients is more significant than in male patients, which may be because women tend to bear more external pressure from family and marriage, which is more likely to induce anxiety. Compared with urban household registration, patients with rural household registration are also more prone to anxiety, and rural household registration is generally under more tremendous economic pressure. Especially elderly patients in rural areas are mostly family laborers who lose their economic source after falling ill. Their children bear the treatment costs, so they are more prone to anxiety due to tremendous psychological pressure. Restless legs syndrome is an adverse reaction that may lead to the increased treatment cost and prolonged treatment time, causing an economic burden to patients and increasing their

anxiety. Therefore, in clinical work, it is necessary to observe whether patients have adverse reactions closely, take relevant measures timely, and pay more attention to rural families and female patients. Nurses should give appropriate psychological counseling to patients with obvious adverse emotions. In this study, the proportion of single, rural household registration, worsening economic situation, fatigue, insomnia, and vascular pain in the depression group were more significant than those in the non-depressed group, having statistically significant differences ($P < 0.05$). Logistic regression analysis showed that rural household registration, worsening economic status, fatigue, insomnia, and vascular pain were independent factors of risk for depression in MHD patients ($P < 0.05$). The economic burden of patients with rural household registration and worsening economic situation is more remarkable, which can seriously affect the mood of patients and induce anxiety and depression. Long-term fatigue and insomnia seriously influence the patients both physically and mentally. Fatigue can be manifested in mental and physical fatigue. After dialysis, patients are often in a state of mental and physical fatigue and are not interested in participating in any activities. Mental fatigue can affect memory, cause unpleasant feelings, and ultimately lead to depression. Vascular pain is an adverse reaction that brings physical pain to the patient and affects the patient's mental health to a certain extent, so patients with vascular pain are prone to depression. This suggests that clinical attention should be paid to rural household registration, worsening economic situations, fatigue, insomnia, and vascular pain to reduce the risk of depression. Future studies may provide further insight by examining interaction effects and conducting regression models factoring in potential confounding variables to give a more comprehensive understanding of the relationships explored in this study.

## CONCLUSIONS

Serum BDNF, NT-3, and 5-HT levels substantially correlate with anxiety and depression in MHD patients. Female, rural household registration, and restless legs syndrome are independent variables of risk for anxiety in MHD patients. Rural household registration, worsening economic status, fatigue, insomnia, and vascular pain are independent variables of risk for depression in MHD patients. The clinical implication of these findings suggests that these indexes may perhaps serve as biological indicators of anxiety and depression amongst patients undergoing MHD. Such investigation can hence contribute to early detection, monitoring, and potentially enable the depiction of novel therapeutic strategies for managing these adverse states.

### Funding

This research was supported by the Mandatory Projects of Nantong Science and Technology Plan–Study on the relationship between depression, anxiety and neurotrophic factors in maintenance hemodialysis patients (MS12021041). The funders had no role in study design, data collection and analysis, decision to publish, or preparation of the manuscript.

## Grant Disclosures

The following grant information was disclosed by the authors:
Mandatory Projects of Nantong Science and Technology Plan–Study: MS12021041.

## Competing Interests

The authors declare there are no competing interests.

## Author Contributions

- Xiaoyan Peng conceived and designed the experiments, performed the experiments, authored or reviewed drafts of the article, and approved the final draft.
- Sujuan Feng performed the experiments, analyzed the data, prepared figures and/or tables, and approved the final draft.
- Poxuan Zhang conceived and designed the experiments, performed the experiments, authored or reviewed drafts of the article, and approved the final draft.
- Shengmei Sang analyzed the data, prepared figures and/or tables, authored or reviewed drafts of the article, and approved the final draft.
- Yi Zhang conceived and designed the experiments, analyzed the data, prepared figures and/or tables, and approved the final draft.

## Human Ethics

The following information was supplied relating to ethical approvals (*i.e.*, approving body and any reference numbers):

All samples obtained in this study were approved by the ethics committee of the Second Affiliated Hospital of Nantong University (First People's Hospital) and abided by the ethical guidelines of the Declaration of Helsinki.

## Data Availability

The raw data is available in the Supplementary File.

## Supplemental Information

Supplemental information for this article can be found online at http://dx.doi.org/10.7717/peerj.16068#supplemental-information.

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
