# Peer review of "Analysis of influencing factors of anxiety and depression in maintenance hemodialysis patients and its correlation with BDNF, NT-3 and 5-HT levels"

_PeerJ, doi:10.7717/peerj.16068_

## Round 0.1 · original submission · Minor Revisions

Please respond and make appropriate revisions based on the reviewers' suggestions and my comments (below). This will greatly improve the quality of the manuscript.

The study investigated the factors contributing to anxiety and depression in maintenance hemodialysis (MHD) patients and their correlation with serum levels of brain-derived neurotrophic factor (BDNF), neurotrophin-3 (NT-3), and serotonin (5-HT). The authors enrolled 120 MHD patients and 60 healthy adults as the control group. The results showed a significant association between anxiety and depression scores and serum levels of BDNF, NT-3, and 5-HT in MHD patients. The study also identified several risk factors for anxiety and depression in MHD patients.

Issues to be revised or improved:
1. Introduction: The rationale for studying the association between BDNF, NT-3, and 5-HT levels with anxiety and depression should be explained in more detail. Provide a strong justification for selecting these specific biomarkers.
2. Methods:
a. Provide more information regarding the selection criteria for MHD patients and the control group. Clearly specify any inclusion or exclusion criteria used.
b. Describe the methods used for assessing anxiety and depression in MHD patients. Include details about the specific assessment tools or questionnaires employed.
c. Clarify the methods used for measuring serum levels of BDNF, NT-3, and 5-HT. Provide information on the reliability and validity of these measurements.
3. Detailed P-values should be shown in Figures 3 and 4.
4. In section 3.4, specify the direction of the correlations between serum NT-3 and 5-HT levels and the SAS score. For example, mention that serum NT-3 and 5-HT levels positively correlated with the SAS score. Similarly, clarify the negative correlation between serum BDNF level and the SDS score.
5. In sections 3.6 and 3.8, include the odds ratios and their corresponding confidence intervals for the independent variables (gender, household registration, monthly dialysis times, insomnia, restless leg syndrome) in the logistic regression model. This will provide a clearer understanding of their impact on anxiety in MHD patients.
6. What is the clinical value and importance of this study? This needs to be added to the Abstract and Conclusion sections.

Reviewer 1 ·

Basic reporting

The manuscript is well-structured and provides valuable insights into the factors associated with anxiety and depression in MHD patients.

Experimental design

1. Provide additional justification for the sample size: Explain the rationale behind selecting a sample size of 120 MHD patients and 60 healthy volunteers. Justify why this sample size is appropriate for the study and whether any power analysis was conducted.
2. Clarify the treatment protocol for MHD patients: Specify the frequency and duration of MHD treatment sessions for the patients included in the study. Provide information about any specific guidelines or protocols followed during the MHD treatment.
3. Specify how the control group was selected: Describe the process used to select the 60 healthy volunteers for the control group. Provide information on how they were matched or controlled for relevant demographic variables such as age and gender.
4. Why this study excluded patients with malignant tumors and hematological diseases?
5. Elaborate on the requirements for case data completeness: Explain what criteria were used to assess the completeness of case data and why it was necessary to exclude incomplete cases. Why or which condition can lead an incomplete case data appear in a hospital?

Validity of the findings

1. In the methods part, authors should clarify the sample size to establish the representativeness and generalizability of the findings.
2. Use consistent terminology: Ensure consistent use of terminology throughout the results section. For example, use "anxiety" throughout instead of switching to "anxiousness." Similarly, use "depression" instead of alternating between "depression" and "depressive symptoms."
3. Specify statistical tests or methods: Mention the statistical tests or methods used to compare the different groups, such as ANOVA or t-tests. This will provide transparency and allow readers to evaluate the appropriateness of the statistical analyses.
4. Suggest follow-up analyses: If relevant, suggest additional analyses that could provide a more comprehensive understanding of the relationships between variables. For example, examining the interaction effects or conducting regression models with potential confounding variables.

Additional comments

1. Provide a brief background on renal failure: The introduction should briefly explain the prevalence and impact of renal failure, including statistics or significant findings, to help set the context for the study.
2. Expand on the impact of ESRD: Further elaborate on how ESRD affects the regular operation of various body organs to highlight the seriousness of the condition.

Reviewer 2 ·

Basic reporting

A more detailed description of the measurement tools used to assess anxiety and depression in MHD patients will strengthen the study.

Experimental design

1. Provide information about the reliability and validity of SAS and SDS scales: Discuss the psychometric properties of the Self-rating Anxiety Scale (SAS) and Self-Rating Depression Scale (SDS). Include details about their reliability and validity, citing publications or studies supporting their use in assessing anxiety and depression.
2. Include information on the reliability of ELISA measurements: Add details about the reliability of the ELISA method used to determine serum BDNF, NT-3, and 5-HT levels. Discuss any measures taken to ensure accuracy and reliability in the laboratory procedures.
3. Clarify how the general data will be compared and analyzed: Provide more specific details on how the general data, such as demographic and clinical characteristics, will be compared between groups. Specify the statistical tests or methods that will be employed to analyze the data.

Validity of the findings

4. Provide p-values: Include the p-values for each statistical test to indicate the significance of the findings. This will help readers assess the strength of the evidence supporting the reported results.
5. Specify the direction of correlations: Clarify whether the correlations are positive or negative. For example, specify whether a positive correlation means that higher anxiety scores are associated with higher NT-3 and 5-HT levels.
6. Provide more context in correlation analysis: Discuss the strength and direction of the correlations observed between variables. This will help readers understand the relationships and their potential implications.
7. Include certainty measures: Consider reporting confidence intervals or standard errors alongside the point estimates to indicate the precision of the results. This will provide a better understanding of the variability of the estimates.

Additional comments

8. Provide rationale for investigating BDNF, NT-3, and 5-HT: Explain the scientific rationale or evidence supporting the investigation of these specific neurotrophic factors (BDNF, NT-3) and 5-hydroxytryptamine (5-HT) in relation to anxiety and depression in MHD patients.

Reviewer 3 ·

Basic reporting

The manuscript presents a study investigating anxiety and depression in MHD patients. The content is expressed in clear and technically correct English, allowing for easy comprehension of the research. The introduction and background provide a context for the study, highlighting its relevance in the broader field of knowledge. The statistical analysis section should be expanded to include explanations of the specific methods employed, adjustments for confounders, and consideration of multiple comparisons. The presentation of tables needs improvement, including clearer headings, labeling, and units of measurement.

Experimental design

a. Expand on the examination of risk variables for anxiety and depression: Explain what factors or variables will be considered as potential risk factors for anxiety and depression in MHD patients, and why. Provide a clear rationale for examining these risk variables and their potential significance.
b. Include information on data preprocessing: Describe any preprocessing steps or data transformations performed on the collected data before conducting the statistical analyses. This might include data cleaning, normality checks, or outlier removal.
c. Consider including information on effect size estimation: Discuss the estimation of effect sizes for the main comparisons and correlations. This will provide a deeper understanding of the practical significance of the findings beyond statistical significance.
d. Did study used any method to limit potential selection bias, or confounding variables. This will provide a more balanced evaluation of the study's findings.

Validity of the findings

e. Consider adjustment for multiple comparisons: If multiple statistical tests were conducted, such as in the univariate analysis, consider adjusting the significance threshold (e.g., Bonferroni correction) to avoid inflated Type I error rates.
f. Expand on the statistical analyses: Provide more details about the statistical methods used in univariate and multivariate analyses. Specify the types of regression models (e.g., logistic regression) and mention any adjustments made for potential confounders.
g. Describe the recruitment process and sampling method: Provide details about how the MHD patients were recruited and the sampling method employed. This information will help readers evaluate the representativeness of the sample and the potential for selection bias.

Additional comments

h. Cite specific references when referring to prior studies: Instead of using general references like "[3-4]", provide the specific citations for the studies that have documented the psychological pressures, anxiety, and depression experienced by MHD patients.
i. Explain the relevance of NT-3 and 5-HT to the study: Elaborate on why NT-3 and 5-HT are being included as factors of interest in the investigation and how they are related to anxiety and depression in MHD patients.

---

## Round 0.2 · accepted · Accept

In carefully evaluating the content of this revised paper, I was satisfied with the responses and revisions made by the authors. The Reviewer's and my concerns have been well addressed. With the necessary revisions and improvements, the quality of this paper has been significantly improved. I believe that this revised manuscript is ready to be considered for publication in this journal.